# Drug Discovery of New Anti-Inflammatory Compounds by Targeting Cyclooxygenases

**DOI:** 10.3390/ph15030282

**Published:** 2022-02-24

**Authors:** Shady Burayk, Kentaro Oh-hashi, Mahmoud Kandeel

**Affiliations:** 1Department of Biomedical Sciences, College of Veterinary Medicine, King Faisal University, Al-Hofuf 31982, Saudi Arabia; drshady1@live.com; 2Department of Chemistry and Biomolecular Science, Faculty of Engineering, Gifu University, 1-1 Yanagido, Gifu 501-1193, Japan; oohashi@gifu-u.ac.jp; 3Department of Pharmacology, Faculty of Veterinary Medicine, KafrelShaikh University, Kafr El-Shaikh 33516, Egypt

**Keywords:** docking, non-steroidal anti-inflammatory drugs, drug discovery, lipoxygenase, cyclooxygenase

## Abstract

The goal of achieving anti-inflammatory efficacy with the fewest possible adverse effects through selective COX-2 inhibition is still being investigated in order to develop drugs with safe profiles. This work shows the efficacy and safety profile of two novel benzimidazole piperidine and phenoxy pyridine derivatives in reaching this goal, which would be considered a major achievement in inflammatory therapy. The compounds were evaluated by virtual screening campaign, in vitro cyclooxygenase 1 and 2 (COX-1 and COX-2) inhibition, in vivo carrageenan-induced rat paw edema assay, cytotoxicity against Raw264.7 cells, and histopathological examination of rat paw and stomach. Two new compounds, compound **1** ([(2-{[3-(4-methyl-1H-benzimidazol-2-yl)piperidin-1-yl]carbonyl}phenyl)amino]acetic acid) and compound **2** (ethyl 1-(5-cyano-2-hydroxyphenyl)-4-oxo-5-phenoxy-1,4-dihydropyridine-3-carboxylate) showed high selectivity against COX-2, favourable drug-likeness and ADME descriptors, a lack of cytotoxicity, relived paw edema, and inflammation without noticeable side effects on the stomach. These two compounds are promising new NSAIDs.

## 1. Introduction

Inflammation is a chain of physiological responses involving a wide range of different molecules and cellular responses [1]. The mediators that emerge from two routes are particularly interesting from the perspective of drug research. Leukotrienes (LTs) are the first class of mediators that contribute to the inflammatory process, and they play a significant part in the inflammatory process overall. LTs are produced by the enzyme Arachidonate 5-lipoxygenase (5-LOX) [2,3]. The second set is more diverse and comes in effect after the activity of COXs that convert arachidonic acid to prostaglandin. PGE2 and PGI2 increase blood flow in inflamed areas by their potent vasodilator action. Prostacyclin (PGI2) is responsible for platelet aggregation and vascular endothelium inhibition. The vasodilation effect of Prostacyclin E2 and Prostacyclin I2 contributes to the protection of the gastric mucosa by boosting mucus secretion and preventing an increase in acidity and pepsin content in the stomach. In kidneys, PGE and PGI play roles in increasing the flow of the blood and regulation of the glomerular filtration rate [4,5].

Nonsteroidal anti-inflammatory drugs (NSAIDs) are a type of prescription that, when taken in sufficient amounts, can reduce pain, inflammation, blood clotting, and fever. NSAIDs may cause several side effects in patients, however, the most dangerous side effects include gastrointestinal ulcers, kidney problems, heart attacks, and bleeding [6]. Despite a few side effects, the drug’s benefits outweigh the risks, making it a viable therapeutic option. NSAIDs are the gold standard in the treatment of inflammatory episodes due to their ability to block the arachidonic acid pathways [7]. NSAIDs act by selective or non-selective inhibition of both enzymes COX-1and COX-2 [8,9]. In addition to lowering inflammation, non-selective medicines work by inhibiting the aggregation of platelets. Additionally, the medication raises the risk of stomach ulcers and bleeding. Gastrointestinal side effects are a prominent concern associated with the use of NSAIDs. These medications cause both direct and indirect irritation of the digestive tract. The acidic components of NSAIDs produce direct irritation of the stomach mucosa as well as inhibit COX-1, resulting in a decrease in prostaglandin levels. Prostaglandin secretion is suppressed throughout the gastrointestinal tract, which increases the formation of stomach acid and decreases the synthesis of mucus, bicarbonates, and the tropical effects of mucosa and epithelial cells [10]. To some extent, all NSAIDs can impair neutrophil activity, but indomethacin, piroxicam, ibuprofen, and all salicylates have the most dramatic effect [11]. There are fewer side effects with selective COX-2 inhibitors, and their gastrointestinal ulcerogenic properties are decreased as well [12]. The pursuit for a less toxic NSAID resulted in the release of two selective COX-2 inhibitors in 1999, celecoxib and rofecoxib [11]. No NSAID is a complete COX-1 or COX-2 inhibitor; rather, each enzyme is inhibited to varying degrees. Given the tremendous load posed by NSAIDs, COX-2 inhibitors appear to be a viable alternative for reducing the risk of major GI bleeding [11].

Therapeutic development will continue to rely extensively on computational tools to aid in the identification and validation of potential drug targets, which will help to accelerate the process [13,14,15,16]. The field of bioinformatics is at the forefront of our efforts to better comprehend the workings of the cell. In order to produce hypotheses for testing, it is necessary to integrate enormous volumes of knowledge from numerous subdisciplines within molecular and cell biology into a cellular model that can be tested. We used such a compilation of methods to deliver new NSAIDs. The selection of such novel compounds was based on their general high selectivity on COX-2, ability to pass drug-likeness and ADME evaluations, strong anti-inflammatory activity, no stomach side effects, and safety when tested in human cell culture.

## 2. Results

### 2.1. Molecular Docking of Compounds

The rationale behind this study was to discover new selective COX-2 inhibitors utilizing virtual screening of a library of chemical compounds that consisted of 1.2 million compounds from the Chembridge Inc core library. Following a two-step, virtual screening procedure that included standard precision docking (SP) followed by extreme precision docking (XP), the compounds with the highest docking scores were obtained from the database (Figure 1). The compound docking scores and ligand efficiency are displayed in Table 1. The compounds are accompanied by favourable hydrogen bonds and lipophilic scores as indicated by the negative values of Glide lipo and Hbond.

By analyzing the ligand interactions of the docked compounds with COX-2 structure, several interaction forces maintained the complexes binding with COX-2, comprising salt bridges, stacking interactions, and hydrogen bonds (Figure 2). Compound **1** formed a hydrogen bond and salt bridge with ARG120. Compounds **2**, **3**, and **6** formed salt bridges with ARG120. Compound **4** formed a stacking interaction with TYR355 and a hydrogen bond with SER530. Compound **5** formed a stacking interaction with TYR385 and salt bridge with SER530. Compound **6** formed a hydrogen bond with ARG120. Compound **7** formed a hydrogen bond with SER530. Collectively, salt bridges with ARG120, hydrogen bonds with TYR335 and SER530, or stacking interactions with TRY385 were the major interaction of the compounds with COX-2. ARG120 and TYR355 are two major residues at the construction site of COX-2. The projected selectivity of compounds against COX-2 can be concluded from the interaction with the side pocket of COX-2 composed of ARG513, ALA516, ILE517, PHE518, VAL523, and LEU531.

### 2.2. ADME Pharmacokinetic Properties and Drug-Likeness Descriptors

Table 2 shows the predicted ADME parameters for the top seven compounds. With no infractions, all compounds passed Lipinski’s rule of five. All compounds showed favourable mw, donor hydrogen bonds, acceptor hydrogen bonds, human oral absorption % (>30%), predicted IC50 value for blockage of HERG K+ channels (<5), absorption from skin (QPlogKp), intestinal absorption, crossing the blood–brain barrier, water accessible surface area, as well as its hydrophilic components. A few minor violations of the prediction parameters were also observed.

### 2.3. Inhibition of COXs

Within the examined seven compounds, compounds **1**–**3** showed the highest selectivity index. Compounds **1**–**3** inhibited the enzymes with compound **1** IC_50_ values of 11.68 ± 1.2 µM and 0.068 ± 0.008 for COX-1 and COX-2, respectively (Table 3). The IC_50_ for compound **2** was 12.22 ± 1.1 and 0.048 ± 0.002 µM for COX-1 and COX-2, respectively. The IC_50_ for compound **3** was 11.11 ± 1.1 and 0.06 ± 0.003 µM for COX-1 and COX-2, respectively. The selectivity index for compound **1** was 124.5 and 68.7-fold higher than indomethacin and diclofenac, respectively. In contrast, it was 1.9-fold lower than celecoxib. Compound **2** has a selectivity index of 254.5, which is higher than compound **1** and makes it the most selective COX-2 inhibitor. Compound **3** was placed second on the selectivity index, with an SI value of 194.9.

### 2.4. The Anti-Inflammatory Assay (Carrageenan-Induced Paw Edema in Rats)

The carrageenan-induced paw edema model was used to evaluate the compounds’ anti-inflammatory activity. Only carrageenan was given to the control animals. The treatment effect was tracked by observing the improvement of rat paw edema in response to drug administration. Carrageenan injections caused rat paws to grow from 0.3 cm to 0.8 cm in size. After 1 h of dosing, diclofenac and indomethacin dramatically reduced rat paw edema (Figure 3). Compounds **1**, **2**, and **3** significantly reduced rat paw size, comparable to diclofenac and indomethacin. The size of the rat paw edema (Figure 4) decreased significantly 30 min after the compounds and drugs were administered. The growth of the rat paw progressed in a substantially comparable manner 3–4 h after injection across all chemicals and medications examined.

### 2.5. Cytotoxicity on Mouse Macrophage-Like Cell Line (Raw264.7 Cells)

Raw264.7 cells were used to test the effects of compounds **1**, **2**, and **3** on cell viability and proliferation. Raw264.7 cells were treated for 24 h in a 96-well plate with compounds (1 µM) in the presence or absence of LPS (1 ug/mL). WST1 reagent was added to each well after treatment and incubated for an additional 2 h. Following that, the OD450/620 ratio was determined. LPS treatment had a significant impact on cell morphology and proliferation. However, all compounds had no effect on WST1 values or cell morphology in Raw264.7 cells. Raw264.7 cells in a 96-well plate were treated with our compounds (1 uM) in the presence or absence of LPS (1 ug/mL) for 24 h. After treatment, the WST1 reagent was added into each well and incubated for an additional 2 h. After that, OD450/620 was measured.

### 2.6. Histopathological Examination of Rat Paw

The control group sections of rat paw displayed a healthy paw tissue structure, exhibiting a typical intact epidermal layer of stratified squamous epithelium and dermal layer with ample capillaries and connective tissue cells (Figure 5A,B). Conversely, the carrageenan-injected paw tissue section examination showed substantial histopathological changes. There was increased dermis thickness attributable to edema with a marked inflammatory cell invasion of the deep dermis compared to the control group (Figure 5C,D). However, the compound **1**-treated group exhibited marked improvement in the degree of edema and the inflammatory cell infiltration (Figure 5E,F). Interestingly, compound **2** exerted a marked anti-inflammatory effect and a pronounced decline in edema and inflammatory cell invasion (Figure 5G,H), which is almost similar to the impact of indomethacin; nevertheless, enhanced improvement in the degree of edema could be seen in the presence of compound **2**. Compound **3** showed weaker action than compounds **1** and **2** manifested by a moderate increase in cell infiltration and edema (Figure 5I,J).

The degree of collagen deposition was investigated by Masson’s stain as a qualitative indicator. Blue-green stained collagen was used for the purpose of assessing collagen deposition progression; the cytoplasm, red blood cells, and muscle were stained red. The density of the blue-green collagen is consistent with the relative amount of deposited total collagen fiber, which represents collagen synthesis, degradation, and remodeling. As shown by Masson’s stain, there were more collagen fibers in the carrageenan group compared with the normal group (Figure 6A). Besides, among the treated groups, the collagen deposition was almost similar to the control nontreated group, except for compound **3**. The quantity of collagen formation was the lowest in the compound **2**-treated group and was better than indomethacin (Figure 6B–E).

### 2.7. Histopathological Examination of Rat Stomach

The control group stomach sections displayed a healthy gastric tissue structure, exhibiting typical intact villi of gastric mucosa with no signs of hemorrhages or congestion; also there is no exfoliation in the mucosal epithelium or inflammatory cell infiltration (Figure 7A,B). Interestingly, compound **1**, compound **2**, compound **3**, and indomethacin-treated groups did not show apparent histopathological changes (Figure 7C–J). The collagen deposition was almost similar to control nontreated animals in compounds **1** and ***2*** (Figure 8A–F). Conversely, compound **3** and indomethacin exerted a mild increase in collagen deposition (Figure 8G–J).

## 3. Discussion

Despite the presence of several NSAIDs, the development of new NSAIDs is attractive, owing to the diverse inflammatory conditions and the need for optimized application in relation to disease and patient conditions. The recent development of NSAIDs comprises modified methyl sulphonyl [17]; pyrimidine-5-carbonitrile hybrids [18], pyrazole derivatives [19], indanone containing spiroisoxazoline derivatives [20], triazole, and oxadiazole compounds [21,22]. Despite the growing repository of newly synthesized NSAIDs, the development of new compounds is strictly required due to the side effects of the currently approved NSAIDs. For instance, even in short-term applications, a high dose of ibuprofen resulted in jejunal perforations [23]; naproxen affects bowel and jejunal integrity [24]. Within days of intake, diclofenac has been proven to raise the risk of heart attacks and strokes by 50% [25].

The intricate interplay between injured tissues and inflammatory cells, which results in the release of inflammatory mediators such as interleukins, necrotic factors, and enzymes such as cyclooxygenases and lipoxygenases, is a crucial aspect of inflammation [26]. The overexpression of COX-1 and COX-2 during inflammatory processes is crucial for the inflammatory signaling system [27]. Controlling inflammation can prevent excessive injury from an aggressive immune response or the development of chronic illness [28].

Drug molecules meet several membrane barriers on their kinetic voyage through the body, including the blood–brain barrier, gastrointestinal epithelial cells, and the membrane target cell. Predicting permeability behind barriers will aid in the prediction of pharmacokinetic parameters and the knowledge of chemical behavior within the body. QikProp estimates and predicts molecular characteristics and provides results for ADME (absorption, distribution, metabolism, and excretion) [29]. Lipinski’s “Rule of Five” and the “Jorgensen Rule-of-Three” were used to determine drug-likeness. The “Jorgensen Rule-of-Three” is based on the qualities of over 90% of 1700 oral medications and meets the “rule of five”, indicating that these molecules can be utilized as potent drugs/inhibitors [30]. The basic drug-likeness and ADME qualities of compounds **1**–**3** indicated generally acceptable parameters, particularly the absence of breaches of Lipinski’s rule of five, good oral absorption, and lack of carcinogenic potential. This was further demonstrated experimentally by the absence of any harmful effect on Raw264.7 cells.

Compounds **1**–**3** were compared to selective and nonselective COX inhibitors for cyclooxygenases inhibition. Compounds **1**–**3** inhibited COX-1 less well than indomethacin and diclofenac but were comparable to the selective medicines celecoxib and rofecoxib. When compared to the nonselective medication indomethacin, compounds **1**–**3** had at least a 100-fold lower impact on COX-1. Furthermore, all compounds inhibited COX-2 at lower concentrations than indomethacin and diclofenac, particularly compound 2, which was comparable to the selective drug celecoxib. The overall selectivity index was in the following order: rofecoxib < celecoxib < compound **2** <compound **3** < compound **1** < diclofenac < indomethacin. This discovery lends credence to the usage of chemicals 1 and 2 as anti-inflammatory medicines. They have no adverse effects on the gastric mucosa and may have no negative effects on the body due to their high selectivity for COX-2 inhibition. Because of their anti-inflammatory efficacy and low gastrointestinal toxicity, COX-2 selective medications have become the most common anti-inflammatory therapy [31]. However, it was quickly discovered that long-term use of COXIBs in arthritic patients was linked to cardiac adverse effects [32]. This has resulted in the withdrawal of some COXIBs, such as rofecoxib, and a warning label for usage in patients with cardiovascular difficulties for other COXIBs, like celecoxib [33]. In this context, the strong selectivity for COX-2 could be linked to such results. In our investigation, compounds **1**–**3** showed medium selectivity between highly selective COXIBs and nonselective COX inhibitors. Because of their lower selectivity index than celecoxib and rofecoxib, these compounds may have an advantage because they have been proven to be kinder on the gastric mucosa and are anticipated to have a lower impact on the cardiovascular system.

In this work, the anti-inflammatory properties of the compounds were evaluated using the carrageenan-induced rat paw swelling test. Carrageenan stimulates exudate release and edema by increasing COX-2 expression, making this model excellent for testing COX-2 inhibitors [28]. The acute carrageenan-induced inflammation in the rat paw corresponds to the findings of in vitro enzymatic tests. Rat paw edema was significantly reduced after treatment with compounds **1**–**3**, indomethacin, and celecoxib. Edema in the paws of rats has two stages. The first stage lasts 2 h and includes histamine and serotonin release. During the second stage, inflammatory exudate and enzymes such as cyclooxygenases are released, and cells are invaded [34]. As a result, the action of compounds **1**–**3** after 1–4 h of inflammation is consistent with cyclooxygenase inhibitory function. Compounds **1** and **2** had a better gastrointestinal safety profile than indomethacin because they did not promote degenerative alterations and exhibited normal collagen deposition in treated rats after a single high dosage. This is consistent with the in vitro COX-1/COX-2 assay’s finding of COX-2 selectivity rather than COX-1. Compounds **1** and **2** appear to be potential NSAIDs based on the entire analysis. Compound **3** was found to be less effective in alleviating inflammatory cells and rat paw edema, as well as increased collagen deposition in the rat stomach.

## 4. Materials and Methods

### 4.1. Software, Chemicals and Kits

The Schrodinger Maestro molecular modeling package (Schrodinger LLC, New York, NY, USA) was used in all virtual screening modeling steps. QikProp tools are an accurate, rapid, and simple-to-use method for predicting molecular properties. GraphPad Prism 7 software was purchased from (GraphPad Software, Inc., San Diego, CA, USA). COX-1 and COX-2 inhibitor screening assay kits were obtained from Cayman Chemical (Ann Arbor, MI, USA). Indomethacin, celecoxib, carboxymethyl cellulose (NaCMC), λ-carrageenan, tris(hydroxymethyl) aminomethane, diclofenac sodium, EDTA, hematin, phenol, and hydrochloric acid were purchased from Sigma-Aldrich (St. Louis, MO, USA). Vernier calipers were manufactured by SMIEC (Shanghai, China). All compounds tested in this work were purchased from ChemBridge Inc. (La Jolla, CA, USA).

### 4.2. Preparation of COX-2 Structure

The protein data bank website was searched for the COX-2 structural PDB ID 5IKQ. The structure was optimized for virtual screening and docking using the protein preparation module of the Schrodinger Maestro molecular modeling tool. The solution’s crystallographic compounds and water molecules were eliminated. The protein was protonated by adding polar hydrogens, and the structures were optimized and energy was decreased by employing the OPLS2005 force field. The docking grids were created using the Maestro grid-generating module, with the defined ligand-binding cavities in the studied structures serving as a starting point. A 20-nanometer grid was built around the active site of the enzyme.

### 4.3. Virtual Screening

To obtain the best candidates, a two-step docking run using SP followed by XP was performed. An integer value of 0.8 was chosen for Van der Waals radius scaling. Extra precision docking is intended to reduce the possibility of a false-positive outcome. Compounds with high docking scores (−12) were chosen for interaction studies with pocket residues, and their binding characteristics and pocket-filling pattern were visually examined. The docking accuracy was confirmed by the low RMSD after redocking of the co-crystallized ligand.

### 4.4. Drug-Likeness and ADME Pharmacokinetic Properties and Descriptors

Pharmacokinetic parameters such as absorption, distribution, metabolism, and excretion were predicted using Qikprop v4.2 tools. The molecular weight, hydrogen bond donor and acceptor, percent oral absorption in humans, an explanation for the violation of Lipinski’s rule of five, and the variety of rotable bonds were among the descriptions. Expected blood/brain partition coefficient, binding to human serum, aqueous solubility (QPlogS) and octanol/water partition constant (QPlogP o/w), apparent MDCK cell permeability and skin permeability (QPlogKp), the SASA element, expected inhibitory concentration (IC50) for blocking HERG K+ Channels, and colorectal adenocarcinoma (Ca) permeability were estimated. To estimate the carcinogenicity and toxicity risk of chemicals, the preADMET web-based application was used.

### 4.5. Enzyme (COXs) Inhibition Assay

Following virtual screening, in vitro experiments for COX enzyme inhibition were performed on compounds with the highest docking scores (seven compounds). The inhibitory effects of compounds on COX-1 and COX-2 were determined using COX inhibitor screening test kits. The test chemicals’ capacity to inhibit the conversion of arachidonic acid to prostaglandin was determined. In test tubes, 25 mM Tris–HCl, pH 8.0, containing 5 mM EDTA, phenol, and 1 mM hematin, was added. The test compounds were dissolved in dimethyl sulfoxide and added in concentrations ranging from 0.005–200 µM. Dimethyl sulfoxide alone was applied to control test containers. COX-1 or COX-2 enzymes were added to test tubes and preincubated for 10 min at 37 °C. The arachidonic acid substrate was added, and the tubes were incubated at 37 °C for 2 min. The compound immunochemical assay was used to calculate the amount of prostaglandin produced. Three separate experiments were used to calculate the *IC*50 values. The selectivity index was calculated as follows:(1)$$Selectivity\;index=\frac{IC50COX-1}{IC50COX-2}$$

### 4.6. Evaluation of Anti-Inflammatory Actions by Carrageenan Induced Paw Edema in Rats

Following in vitro testing, three compounds with the highest selectivity against COX-2 were chosen for in vivo investigations utilizing the rat paw edema assay. After subcutaneous carrageenan injection into the rodent’s paw, plasma extravasation, tissue proliferation, plasma protein release, and neutrophilic extravasation, all of which are caused by arachidonic acid digestion, produced inflammation. Following the carrageenan infusion, the main phase lasted 2 h. The following process is remodeling, which takes 5 h and begins at the 3rd h. The main stage is distinguished by the appearance of histamine and serotonin. The development of prostaglandins, proteases, and lysosomes causes the second stage of edema. Adult male albino rats weighing 120–150 g were divided into seven groups, each with seven rats, and kept at 25 °C with a 12-h light/dark cycle. Subplantar injection of 0.2 mL of 1% carrageenan solution in 0.9 percent saline into the right hind paw of rats was used to induce swelling and inflammation. To determine the inflammation generated by carrageenan, the Vernier Caliper was used to measure the thickness of rat paws before and after injection. Compounds **1**–**3**, as well as indomethacin and diclofenac sodium, were suspended in 1% NaCMC in normal saline and given orally at 10 mg/kg in a total volume of 1 mL per rat, with the negative control getting 1 mL of 1% NaCMC-saline solution. Paw edema was measured at 0, 0.5, 1, 2, 3, 4, and 5 h after the drug was administered. The edema inhibition was calculated by calculating the difference in thicknesses between the treated and control groups [35,36].

### 4.7. Histopathological Examination

#### 4.7.1. Rat Paw

The rats’ paws were collected and washed in a saline solution after the end of rat paw experiment. Specimens were fixed in 10% natural formalin for 2 days before being washed overnight with running water. The cleaned samples were dehydrated in ethyl alcohol at increasing percentages, starting with 70% and ending with absolute alcohol. The samples were cleaned by immersing them in xylol for 2 h. For 3 h at 37 °C, the cleared samples were placed in a sealed jar containing 50% paraffin in xylol. The samples were then soaked in melting paraffin for 2 h at 48 °C before being blocked in hard paraffin and sliced into 5-micron slices. The sections were stained with hematoxylin and eosin (H&E) and Masson’s trichrome. Sections were mounted with Canada balsam and covered with a coverslip in preparation for histological examination.

#### 4.7.2. Rat Stomach and Evaluation of the Ulcerogenic Effect of Test Compounds

Compounds **1**–**3** were examined for ulcerogenic effects in the rat stomach after showing anti-inflammatory advantages in a rat paw test. Rats were deprived for 24 h at 25 °C with a 12 h light/dark cycle. Indomethacin was used as a control. All compounds were given in a single dose of 20 mg/kg orally. After 24 h, the rats were slaughtered, and their stomachs were removed and washed with normal saline. Binocular magnification was used to inspect each group’s stomachs for the existence of a gastric lesion on the mucosa. Following that, the stomachs were kept in 10% *w*/*v* formalin for histological investigation, as previously described.

### 4.8. In Vitro Cytotoxicity Using Mouse Macrophage-Like Cell Line (Raw264.7 Cells)

Dulbecco’s modified Eagle’s minimum essential medium containing 5% heat-inactivated fetal bovine serum was used to maintain Raw264.7 cells (ATCC). Raw264.7 cells (0.8104 cells/100 µL) in a 96-well plate were treated for 24 h with the indicated compounds at 1 µM or vehicle (DMSO) in the presence or absence of LPS (1 µg/mL) to detect cell viability based on mitochondrial activity. WST-1 solution (Dojindo) was then added to each well and incubated at 37 °C for an additional 2 h. According to a previous study, cell viability was determined using the difference in absorbance at 450 and 620 nm as an indicator [37].

### 4.9. Statistical Analysis

All results were presented as means standard deviations (SD). Statistical analysis was performed as previously described [38]. When appropriate, statistical analysis was carried out using two-way analysis of variance (ANOVA) followed by the Bonferroni’s test, or one-way ANOVA followed by the Tukey’s test. *p* values of less than 0.05 were regarded as statistically significant. GraphPad Prism version 5.00 for Windows (San Diego, CA, USA) was used for all statistical analyses.

## 5. Conclusions

A virtual screening effort of 1.2 million compounds was launched in a study to uncover novel COX-2 selective NSAIDs. A docking score filtration of the compounds offered a hint for future investigation using experimental techniques. Compound **1** ([(2-{[3-(4-methyl-1H-benzimidazol-2-yl)piperidin-1-yl]carbonyl}phenyl)amino]acetic acid) and compound **2** (ethyl 1-(5-cyano-2-hydroxyphenyl)-4-oxo-5-phenoxy-1,4-dihydropyridine-3-carboxylate) showed promising anti-inflammatory activity. Compounds **1** and **2** were highly selective for COX-2, reduced rat paw edema, were non-cytotoxic on cells, decreased inflammatory cells infiltration and edema, and did not display adverse responses on rat stomach. These two compounds are recommended as new promising NSAIDs.

## 6. Patents

Compounds **1** and **2** in this manuscript are subject to patent application at the US patent office. USPTO application no. 17665553 (methods of treating inflammation) and attorney docket no. 32087.20.

## Figures and Tables

**Figure 1 pharmaceuticals-15-00282-f001:**
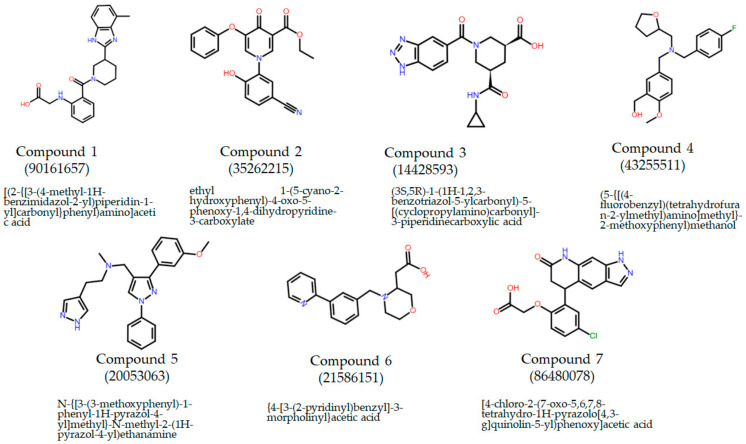
Compounds’ chemical names, structures, and IDs. The compounds were purchased from Chembridge (San Diego, CA, USA).

**Figure 2 pharmaceuticals-15-00282-f002:**
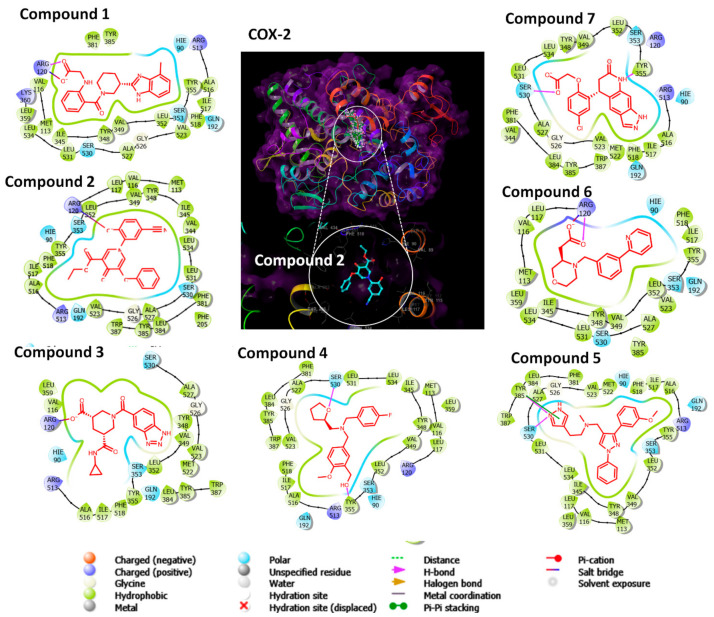
The docking site and ligand interactions of compounds **1**–**7**. The ligand interaction of every compound is provided. Positively charged residues are in blue, while hydrophobic residues are in green. Salt bridges are depicted as double-colored lines in blue and red. Stacking interactions are represented by green lines.

**Figure 3 pharmaceuticals-15-00282-f003:**
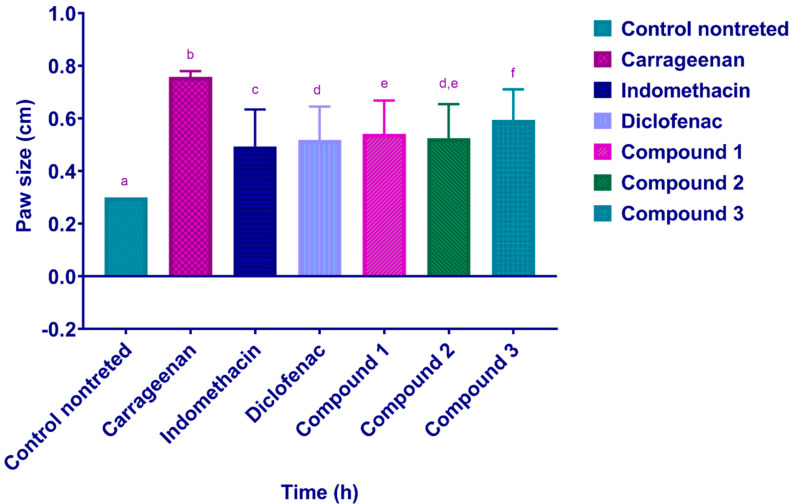
Average rats paw edema size (cm). The control group was treated with 0.2 mL of 1% carrageenan. Compounds **1**, **2**, and **3** were given orally at a dose rate of 10 mg/kg. The control nontreated group received NaCMC. The rat paw size was evaluated for 5 h after treatment. Each column represents the mean ± SD. Mean values in each plot followed by a different lowercase letter (a, b, c, d, e and f) are significantly different at *p* ≤ 0.05.

**Figure 4 pharmaceuticals-15-00282-f004:**
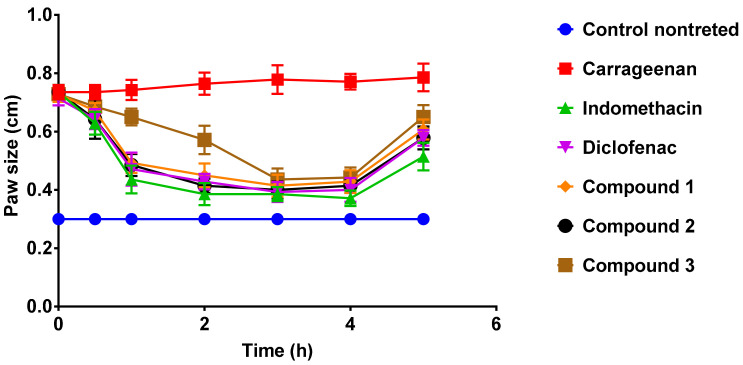
The progress of rat paw edema (cm) for 5 h after compounds administration. The control group was treated with 0.2 mL of 1% carrageenan. Compounds **1**, **2**, and **3** were given orally at a dose rate of 10 mg/kg. The control nontreated group received NaCMC. The measurement was performed at 0, 30 min, 1, 2, 3, 4 and 5 h after administration of the drugs.

**Figure 5 pharmaceuticals-15-00282-f005:**
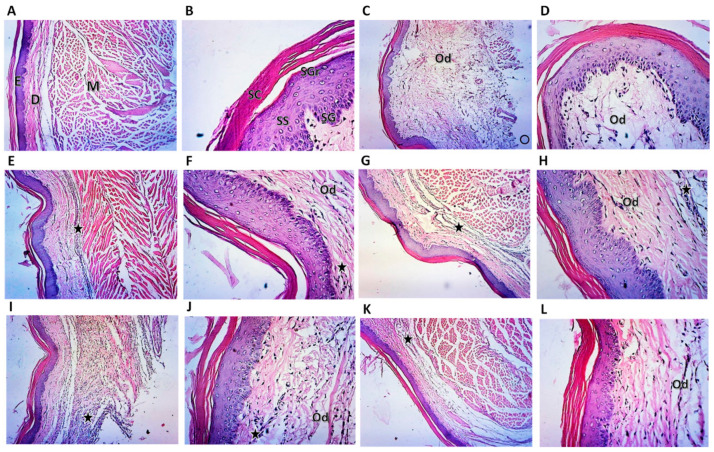
Histopathologic examination of rat paw stained by H&E. (**A**) Photomicrograph of control rat paw at 10× magnification showing normal epidermal layers (E), dermis (D), and muscle layer (M); notice the absence of any signs of an inflammatory reaction. (**B**) Photomicrograph of control rat paw at 40× magnification showing normal SC, stratum corneum; SGr, stratum granulosum; SS, stratum spinosum; and SG, stratum germinativum. (**C**) Photomicrograph of rat paw at 10× magnification of carrageenan-treated group showing marked thickening of the dermal layer, inflammatory reaction in the deep dermis, and wide separation between fibers due to edema (Od). (**D**) Photomicrograph of rat paw at 40× magnification of carrageenan-treated group showing edema (Od). (**E**) Photomicrograph of rat paw at 10× magnification of carrageenan-treated group followed by treatment with compound **1** showing a significant decrease in the dermal thickness, inflammatory reaction in the deep dermis (black star), and moderate improvements in edema (Od). (**F**) Photomicrograph of rat paw at 40× magnification of carrageenan-treated group followed by treatment with compound **1** showing slight inflammatory reaction (black star) and edema (Od). (**G**) Photomicrograph of rat paw at 10× magnification of carrageenan-treated group followed by treatment with compound **2** showing a substantial decrease in both the inflammatory reaction (black star) and edema (Od). (**H**) Photomicrograph of rat paw at 40× magnification of carrageenan-treated group followed by treatment with compound **2** showing slight inflammatory reaction (black star) and edema (Od). (**I**) Photomicrograph of rat paw at 10× magnification of carrageenan-treated group followed by treatment with compound **3** showing a significant decrease in the degree of inflammatory reaction (black star) and a moderate decrease in edema (Od). (**J**) Photomicrograph of rat paw at 40× magnification of carrageenan-treated group followed by treatment with compound **3** showing moderate inflammatory reaction (black star) and edema (Od). (**K**) Photomicrograph of rat paw at 10× magnification of carrageenan-treated group followed by treatment with indomethacin showing a significant decrease in the degree of inflammatory reaction (black star) and a moderate decrease in edema (Od). (**L**) Photomicrograph of rat paw at 40× magnification of carrageenan-treated group followed by treatment with indomethacin showing slight inflammatory reaction (black star) and edema (Od).

**Figure 6 pharmaceuticals-15-00282-f006:**
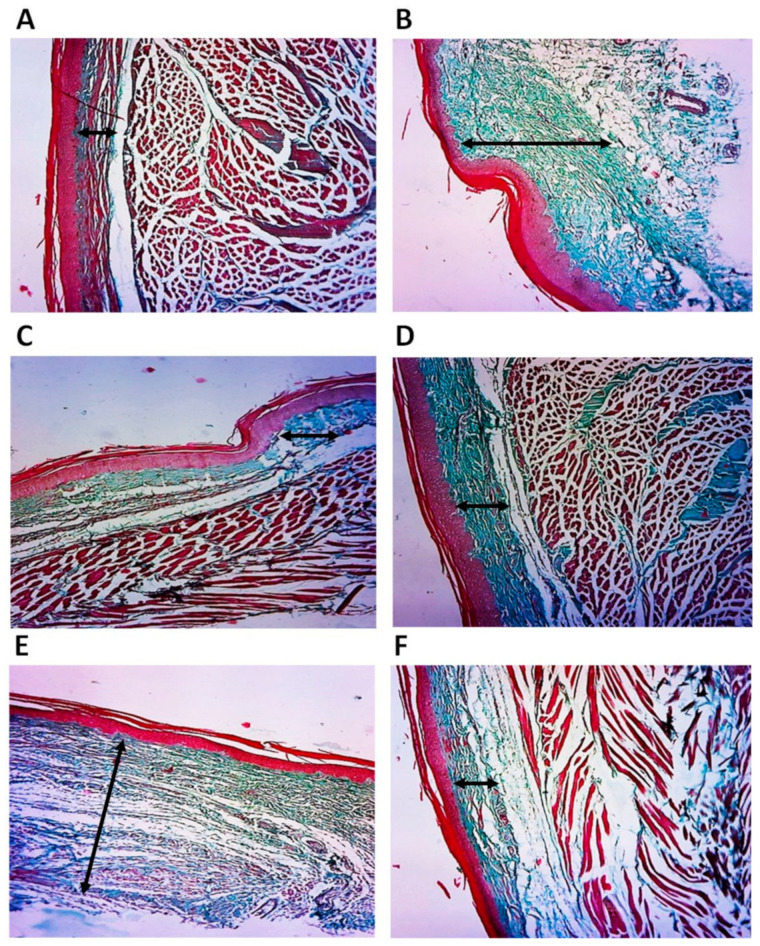
Histopathologic examination of rat paw stained by Masson’s Trichrome stain. Collagen fibrils are stained blue-green. The double-headed arrow represents collagen fibrils thickness between epidermis and dermis. (**A**) Photomicrograph of control rat paw at 10× magnification showing normal collagen deposition. (**B**) Photomicrograph of rat paw at 10× magnification of carrageenan-treated group showing intense collagen deposition. (**C**) Photomicrograph of rat paw at 10× magnification of carrageenan-treated group followed by treatment with compound **1** showing slight collagen deposition (**D**). Photomicrograph of rat paw at 10× magnification of carrageenan-treated group followed by treatment with compound **2** showing slight collagen deposition. (**E**) Photomicrograph of rat paw at 10× magnification of carrageenan-treated group followed by treatment with compound **3** showing significant collagen deposition. (**F**) Photomicrograph of rat paw at 10× magnification of carrageenan-treated group followed by treatment with indomethacin showing slight collagen deposition.

**Figure 7 pharmaceuticals-15-00282-f007:**
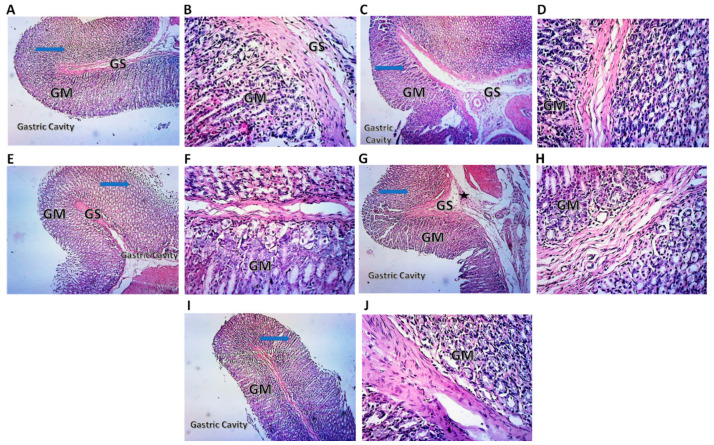
Histopathologic examination of rat stomach stained by H&E. (**A**) Photomicrograph of control rat stomach at 10× magnification showing normal gastric layers. Blue arrow shows intact appearance of histological structure of the epithelium and mucosa layer; GM, Gastric Mucosa; GS, Gastric Submucosa. (**B**) Photomicrograph of control rat stomach at 40× magnification showing absence of any signs of an inflammatory reaction. (**C**) Photomicrograph of rat stomach at 10× magnification of treatment with compound **1** showing normal gastric layers. Blue arrow shows intact appearance of histological structure of the epithelium and mucosa layer; GM, Gastric Mucosa; GS, Gastric Submucosa. (**D**) Photomicrograph of rat stomach at 40× magnification of treatment with compound **1** showing absence of any signs of an inflammatory reaction. (**E**) Photomicrograph of rat stomach at 10× magnification of treatment with compound **2** showing normal gastric layers. Blue arrow shows intact appearance of histological structure of the epithelium and mucosa layer; GM, Gastric Mucosa; GS, Gastric Submucosa. (**F**) Photomicrograph of rat paw at 40× magnification of treatment with compound **2** showing absence of any signs of an inflammatory reaction. (**G**) Photomicrograph of rat stomach at 10× magnification of treatment with compound **3** showing slight degeneration of gastric layers. (**H**) Photomicrograph of rat paw at 40× magnification of treatment with compound **3** showing absence of any signs of an inflammatory reaction. (**I**) Photomicrograph of rat stomach at 10× magnification of treatment with indomethacin, showing slight degeneration of gastric layers. (**J**) Photomicrograph of rat paw at 40× magnification of treatment with indomethacin showing absence of any signs of an inflammatory reaction.

**Figure 8 pharmaceuticals-15-00282-f008:**
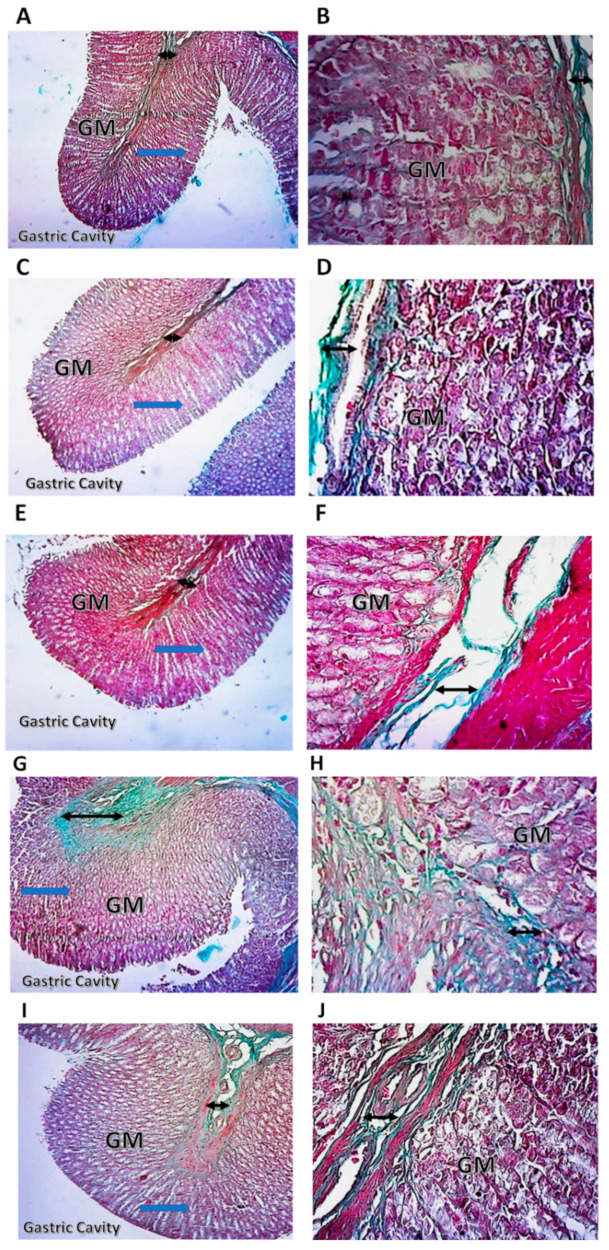
Histopathologic examination of rat stomach stained by Masson’s Trichrome. Collagen fibrils are stained blue-green. The double-headed arrow represents collagen fibrils thickness in the submucosa. (**A**) Photomicrograph of control rat stomach at 10× magnification showing normal collagen deposition. (**B**) Photomicrograph of rat stomach at 40× magnification of control group showing normal collagen deposition. (**C**) Photomicrograph of rat stomach at 10× magnification of treatment with compound **1** showing normal collagen deposition. (**D**) Photomicrograph of rat stomach at 40× magnification of treatment with compound **1** showing normal collagen deposition. (**E**) Photomicrograph of rat stomach at 10× magnification of treatment with compound **2** showing slight collagen deposition. (**F**) Photomicrograph of rat stomach at 40× magnification of treatment with compound **2** showing slight collagen deposition. (**G**) Photomicrograph of rat stomach at 10× magnification of treatment with compound **3** showing significant collagen deposition. (**H**) Photomicrograph of rat stomach at 40× magnification of treatment with compound **3** showing significant collagen deposition. (**I**) Photomicrograph of rat stomach at 10× magnification of treatment with indomethacin showing significant collagen deposition. (**J**) Photomicrograph of rat stomach at 40× magnification of treatment with indomethacin showing collagen deposition.

**Table 1 pharmaceuticals-15-00282-t001:** The obtained compounds with the highest docking score after virtual screening using PDB ID 5IKQ.

Compound #	Docking Score (kcal/mol)	Glide H Bond	Glide Lipo	Glide Ligand Efficiency
**1**	−13.645	−0.908	−3.705	−0.471
**2**	−12.442	−0.172	−5.079	−0.444
**3**	−12.523	−0.73	−3.433	−0.482
**4**	−12.41	0	−5.106	−0.477
**5**	−13.14	−0.87	−5.373	−0.453
**6**	−13.408	0	−3.123	−0.583
**7**	−13.222	−0.813	−4.053	−0.509

**Table 2 pharmaceuticals-15-00282-t002:** Drug-likeness and ADME properties of compounds.

	Acceptable Range	Compound 1	Compound 2	Compound 3	Compound 4	Compound 5	Compound 6	Compound 7
Mol-Mw	130–725	392.457	376.372	357.368	371.779	387.483	359.44	312.368
Donor HB	0–6	2	2	3	3	1	1	1
Accept HB	2–20	6.5	6.45	9.5	6.25	5.25	6.15	6.7
Human oral absorption %	>80 high<30 low	91.392	63.711	41.441	60.987	100	100	61.58
Rule of five	Maximum 4	0	0	0	0	0	0	0
Qplog P o/w	Maximum 3	3.886	1.274	0.168	2.367	4.446	3.65	0.623
QPlogHERG	Concern below −5	−3.629	−4.202	−1.498	−3.234	−7.581	−6.219	−4.404
QPlogKp	−8.0–−0.1	−2.236	−4.012	−5.679	−4.871	−2.982	−2.777	−4.137
QPPCaco	<25 poor>500 great	213.531	43.389	5.69	13.419	442.32	926.454	53.849
QPlogBB	−3.0–1.2	−0.742	−0.713	−2.218	−1.663	−0.232	0.089	−0.395
QPPMDCK	<25 poor>500 great	118.581	23.433	3.263	14.645	226.628	912.502	29.595
QPlogKhsa	−1.5–1.5	0.21	−0.056	−0.83	−0.067	0.77	0.211	−0.253
SASA	300–1000	640.252	624.75	609.578	586.73	712.49	655.337	584.481
FISA	7.0–330.0	112.838	122.221	265.073	239.562	78.779	44.92	112.33
Carcinogenicity in mouse	Negative	Negative	Negative	Negative	Negative	Negative	Negative	Negative
Carcinogenicity in rat	Negative	Negative	Negative	Negative	Negative	Negative	Negative	Negative

**Table 3 pharmaceuticals-15-00282-t003:** The estimated IC50 values (µM) for Compounds **1**–**3** against COX-1 and COX-2. Celecoxib, rofecoxib, indomethacin, and diclofenac were used as control drugs.

	COX-1	COX-2	SI
Celecoxib	14.7 ± 1.045	0.045 ± 0.005	326.6
Rofecoxib	14.5 ± 1.125	0.025 ± 0.005	580
Indomethacin	0.1 ± 0.015	0.0725 ± 0.01	1.38
Diclofenac	0.05 ± 0.006	0.02 ± 0.001	2.5
Compound **1**	11.68 ± 1.2	0.068 ± 0.008	171.8
Compound **2**	12.22 ± 1.1	0.048 ± 0.002	254.5
Compound **3**	11.11 ± 1.1	0.06 ± 0.003	194.91
Compound **4**	7.62 ± 0.05	0.21 ± 0.01	36.3
Compound **5**	10.57 ± 0.33	0.08 ± 0.009	132.64
Compound **6**	11.52 ± 0.13	0.06 ± 0.075	190.88
Compound **7**	9.19 ± 0.046	0.097 ± 0.015	95.068

## Data Availability

Data is contained within the article.

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
