# Peer review of "Drug Discovery of New Anti-Inflammatory Compounds by Targeting Cyclooxygenases"

_pharmaceuticals, 2022, doi:10.3390/ph15030282_

Round 1

Reviewer 1 Report

The study from Burayk et al., presents a great interest since there is a continuous need of anti-inflammatory drugs, inflammation being a central process in many diseases. However prior to be accepted, the manuscript should be significatively improved. Here are some comments. 

 Except, the part 2.6 all the results are too concisely presented and documented. 

1- Section 2.1, please reintroduce the rational behind the virtual screening strategy, targeted binding sites, size of the library ect..Tables 1 and 2 should be inverted, since Table 2 presents the results of the virtual screening. Table 1 should be converted to a figure and SAR explicited. It is not clear what are the structural basis of the inhibition when examining figure 1. The legend of Figure 1 should be better documented and Figure 1 itself should be more analysed in the text. 

2- Section 2.2 I am fully convinced of the conclusions, please rephrase and better analysed, lines 77-78; the Lipinski's rule of five are not always a criterion of effective drugs, since many drugs used in the clinic are really out of the Rule of Five. 

3- Section 2.4, Please introduce the model prior to present results, table 5 is not necessary since you present Figure 2 or replace by curves or histogramms that are more informative. Figure 2 please provide a statistical analysis and give the time of measurement (it is not clear from the legend)

4- Section 2.5, it is not clear why authors used HEK293 since it is not really biologically relevant for their study, please suppress. Moreover, this is not introduced in the Mat and Methods section. 

5- Discussion and conclusion parts are not really well documented, they should be significatively rewritten. 

6- The number and the pertinence of the cited publications should be revised, to better cover the field of NSAID targeting COX and COX inhibitors which are very well documented fields. 

Author Response

Reviewer 1

The study from Burayk et al., presents a great interest since there is a continuous need of anti-inflammatory drugs, inflammation being a central process in many diseases. However prior to be accepted, the manuscript should be significatively improved. Here are some comments. 

 Except, the part 2.6 all the results are too concisely presented and documented. 

  • Section 2.1, please reintroduce the rational behind the virtual screening strategy, targeted binding sites, size of the library ect..Tables 1 and 2 should be inverted, since Table 2 presents the results of the virtual screening. Table 1 should be converted to a figure and SAR explicited. It is not clear what are the structural basis of the inhibition when examining figure 1. The legend of Figure 1 should be better documented and Figure 1 itself should be more analysed in the text. 

Response to reviewer comments

Section 2.1 was modified to add the rationale of the study, the used library and targets.

Table 1 was converted to figure 1 while table 1 was modified by removing the chemical names. Consequently, all figures and tables were renumerated.

The legend of Figure 1 was modified to better document the content of the figure.

Figure 1 was discussed in details in the text.

By analyzing the ligand interactions of the docked compounds with COX-2 structure, several interaction forces maintained the complexes binding with COX-2, comprising salt bridges, stacking interactions and hydrogen bonds (Figure 2). Compounds 1 formed hydron bond and salt bridge with ARG120. Compounds 2, 3 and 6 form salt bridges with ARG120. Compound 4 forms stacking interaction with TYR355 and a hydrogen bond with SER530. Compound 5 formed stacking interaction with TYR385 and salt bridge with SER530. Compound 6 formed hydrogen bond with ARG120. While, compound 7 formed hydrogen bond with SER530. Collectively, salt bridges with ARG120, hydrogen bonds with TYR335 and SER530 or stacking interactions with TRY385 were the major interaction of the compounds with COX-2. ARG120 and TYR355 are two major residues at the constriction site of COX-2. The projected selectivity of compounds against COX-2 ca be concluded from the interaction with the side pocket of COX-2 composed of ARG513, ALA516, ILE517, PHE518, VAL523 and LEU531.

  • Section 2.2 I am fully convinced of the conclusions, please rephrase and better analysed, lines 77-78; the Lipinski's rule of five are not always a criterion of effective drugs, since many drugs used in the clinic are really out of the Rule of Five. 

Response to reviewer comments

The sentence was modified and corrected as follows “With no infractions, all compounds passed Lipinski's rule of five”

  • Section 2.4, Please introduce the model prior to present results, table 5 is not necessary since you present Figure 2 or replace by curves or histogramms that are more informative. Figure 2 please provide a statistical analysis and give the time of measurement (it is not clear from the legend)

Response to reviewer comments

Section 2.4 was modified as instructed

The model was described at the start of section 2.4 as follows “The carrageenan-induced paw edema model was used to evaluate the compounds' anti-inflammatory activity. Only carrageenan was given to the control animals. The treatment effect was tracked by observing the improvement of rat paw edema in response to drug administration. Carrageenan injections caused rat paws to grow from 0.3 cm to 0.8 cm in size.”

We agree with the reviewer's comment that Table 5 is not necessary so it is removed in the revised version.

The statistical analysis was provided in a separate section in the methods. As follows

“All results were presented as means standard deviations (SD). Statistical analysis was performed as previously described [22]. When appropriate, statistical analysis was carried out using two-way analysis of variance (ANOVA) followed by the Bonferroni's test, or one-way ANOVA followed by the Tukey's test. P values of less than 0.05 were regarded as statistically significant. GraphPad Prism version 5.00 for Windows (San Diego, California, USA) was used for all statistical analyses.”

  • Section 2.5, it is not clear why authors used HEK293 since it is not really biologically relevant for their study, please suppress. Moreover, this is not introduced in the Mat and Methods section. 

Response to reviewer comments

We removed the data from HEK293 cells. Instead, we used mouse macrophage-like cell line to assess the effect of compounds on cell morphology and proliferation.

The methods and results sections were modified as follows:

2.5. Cytotoxicity on mouse macrophage-like cell line (Raw264.7 cells)

Raw264.7 cells were used to test the effects of compounds 1, 2, and 3 on cell viability and proliferation. Raw264.7 cells were treated for 24 hours in a 96-well plate with compounds (1 µM) in the presence or absence of LPS (1 ug/ml). WST1 reagent was added to each well after treatment and incubated for an additional 2 hours. Following that, the OD450/620 ratio was determined. LSP treatment had a significant impact on cell morphology and proliferation. However, all compounds had no effect on WST1 values or cell morphology in Raw cells.

4.8. In vitro cytotoxicity using mouse macrophage-like cell line (Raw264.7 cells)

Dulbecco's modified Eagle's minimum essential medium containing 5% heat-inactivated fetal bovine serum was used to maintain Raw264.7 cells. Raw264.7 cells (0.8104 cells/100 µL) in a 96-well plate were treated for 24 hours with the indicated compounds at 1 µM or vehicle (DMSO) in the presence or absence of LPS (1 µg/mL) to detect cell viability based on mitochondrial activity. WST-1 solution (Dojindo) was then added to each well and incubated at 37°C for an additional 2 hours. According to a previous study, cell viability was determined using the difference in absorbance at 450 and 620 nm as an indicator [22].

  • Discussion and conclusion parts are not really well documented, they should be significatively rewritten. 

Response to reviewer comments

The discussion and conclusions are rewritten and supported by evidences and references.

Please refer to the track changes file to tack the modifications.

6- The number and the pertinence of the cited publications should be revised, to better cover the field of NSAID targeting COX and COX inhibitors which are very well documented fields. 

Response to reviewer comments

Thank you very much for this constructive comments.

We enriched the manuscript with more literature details in the introduction and the discussion to better cover the field of NSAID targeting COX and COX inhibitors.

Reviewer 2 Report

The paper „Drug discovery of new anti-inflammatory compounds by targeting cyclooxygenases” is of good quality. After reading this manuscript, I have a few comments. I believe the authors should do Cox inhibition experiments for all 7 substances. The differences in the obtained docking results are especially not so large in this group of compounds. In silico and in vivo experiments often do not produce the same results. It seems to me that Table 2 would be clearer without the names of the substances (they are in Table 1) and the substance numbers in the text should be written in bold.

Author Response

Reviewer 2

The paper „Drug discovery of new anti-inflammatory compounds by targeting cyclooxygenases” is of good quality. After reading this manuscript, I have a few comments. I believe the authors should do Cox inhibition experiments for all 7 substances.

The differences in the obtained docking results are especially not so large in this group of compounds. In silico and in vivo experiments often do not produce the same results.

It seems to me that Table 2 would be clearer without the names of the substances (they are in Table 1) and the substance numbers in the text should be written in bold.

Response to reviewer comments

We appreciate the great feedback from the reviewer.

We're also excited to highlight that we've already performed COX inhibition trials on all of the compounds, but we only provided the top three most selective ones. We had incorporated the details for all seven compounds to this revised version. Please see Table 4 as well as the results section for additional information.

The docking results are really close, and we concur with the reviewer. However, numerous other parameters, such as pKa value, hydrophobicity, and size, impact the drug's activities in vivo. This is where the drug's disposition and tissue penetration are controlled. Also, despite we tested the top selective three compounds, only 2 compounds were approved after in vivo assays as the 3rd one was less effective and more toxic.

The names of the drugs in Table 2 have been omitted.

The numbers for the compounds were bolded.

Round 2

Reviewer 1 Report

The authors have answered properly to most of my comments, thus the manuscript can be accepted for publication in Pharmaceuticals.